# Best practices for implementing biosafety inspections in a clinical laboratory: Evidence from a multi-site experimental study

**Qiaoling Qin[1], Cynthia Tseng[2], Wenlin Chen[1]\*, Chung-Li Tseng[3]**

**1** University of Electronic Science and Technology of China, Chengdu, China, **2** Cornell University, New York, New York, United States of America, **3** University of New South Wales, Sydney, Australia

\* wenlinchen@uestc.edu.cn

**Data Availability Statement:** All relevant data are within the paper and its Supporting information files.

## Abstract

### Objectives

To explore the key components when designing best practice inspection interventions, so as to induce high compliance with safety guidelines for laboratory workers.

### Methods

Five key components of an inspection intervention, identified from a focus group discussion, were used as the attributes of a discrete choice experiment (DCE). In the DCE, participants were presented with two hypothetical scenarios and asked to choose the scenario in which they were more willing to comply with the laboratory safety guidelines. Data were collected from 35 clinical laboratories in seven healthcare institutes located in Chengdu, China. In total, 188 laboratory workers completed the DCE. The collected data were analyzed using conditional logit regression and latent class analysis.

### Results

Five key attributes were identified as the most important ones to best ensure laboratory safety: the inspector, the inspection frequency, the inspection timing, the communication of the inspection outcome, and a follow-up with either a reward or a punishment. By investigating the laboratory workers' responses to the attributes, properly implementing the five attributes could improve the workers' compliance from 25.86% (at the baseline case) to 74.54%. Compliance could be further improved with the consideration of the laboratory workers' heterogeneous reactions. In this study, two classes of workers, A and B, were identified. Compliance percentages for Classes A and B would be improved to 85.48% and 81.84%, respectively, when the key attributes were properly implemented for each class. The employment type and the size of the laboratory could be used to predict class membership.

**Funding:** This work was supported in part by a grant from the National Natural Science Foundation of China (grant number: 71902017) and the China Scholarship Council (grant number: 202206075012). The funders had no role in study design, data collection and analysis, decision to publish, or preparation of the manuscript.

**Competing interests:** The authors have declared that no competing interests exist.

## Conclusion

The findings indicate the importance of an employee-centered approach in encouraging a worker's compliance. This approach also supports the design of tailored interventions by considering the laboratory workers' heterogeneous responses to the interventions.

## Introduction

Clinical laboratories play a crucial role in modern medical care, as about 70% of medical decisions are made, based on laboratory test results [1]. Clinical laboratory workers conduct and analyze diagnostic tests to aid physicians in the diagnosis, treatment, and management of their patients. As their roles involve handling clinical specimens collected from patients, including biochemical, microbiological, hematological, serological, pathological, and cytological tests, the laboratory workers are exposed to hazardous substances, including chemicals and biological agents [2].

Clinical laboratory workers are subject to infectious agents and *laboratory-acquired infections*, increasing the risk of serious occupational hazards [3–5]. For example, literature showed that the probability for clinical laboratory workers to acquire a tuberculosis infection was about seven times higher than that for the general population in the United Kingdom [6]. In addition, the 2014 Ebola outbreak presented another example of high risks in clinical laboratories [7].

Although the importance of safety guidelines has been recognized by clinical laboratory workers, full compliance with the guidelines has not been achieved. For example, one study found that when culturing specimens of patients with suspected brucellosis, a clinical laboratory worker did not adhere to the biosafety procedure by using proper personal protective equipment (PPE), which resulted in an occupational exposure [5]. In addition, many other studies have found that compliance with biosafety guidelines is subpar; despite the laboratory worker's knowledge of the required code of conduct [8–10].

Workplace safety inspection, which is an important intervention mechanism in safety management, has been proved as an essential way of ensuring the safety behavior of workers in a variety of industries, such as the construction industry [11], the coal mining industry [12], and the manufacturing industry [13]. Inspections were also emphasized in the safety guidelines to ensure that a clinical laboratory worker would be compliant with required biosafety guidelines. The purpose of this study, therefore, was to identify the best practice inspection interventions needed to improve clinical laboratory workers' compliance in an effective way. More specifically, this study aimed to (i) identify the key attributes to be considered when designing an inspection intervention; (ii) measure the effect of these attributes on the laboratory workers' intention for compliance with safety guidelines; (iii) examine the effect of heterogeneous responses to the inspection intervention; and (iv) identify best practice inspection interventions.

Our study contributes to the extant literature in several ways. First, we addressed a significant gap in the safety management of clinical laboratories by identifying evidence-based best practices for inspection interventions. Secondly, we adopted an employee-centered approach, rather than a top-down approach, to gather solid evidence for identifying the most suitable interventions. Specifically, these inspection interventions systematically addressed the clinical lab workers' preferences to the interventions. Since the ultimate goal of an intervention is to ensure that lab workers are more willing to comply with biosafety guidelines, understanding

how clinical lab workers respond to interventions is essential and indispensable when designing effective interventions.

## Methods

### Study design

A discrete choice experiment (DCE) was designed to assess the laboratory workers' responses toward the components of an inspection intervention. DCEs are widely used in the healthcare field to assess a participant's responses using hypothetical scenarios, for instance, over the preferred forms of medication. DCEs were also used to investigate a patient's preferences for various treatments [14, 15], the choice of health providers [16, 17], or the choice of healthcare insurance [18, 19]. In Chen and Tseng [20], a DCE was conducted to evaluate the health care workers' (HCWs') responses for hand hygiene interventions.

In a DCE, a hypothetical scenario is characterized by attributes; and each attribute is described by different levels. To identify the relevant attributes and levels, a focus group discussion was conducted with 3 directors and 5 laboratory workers from the West China Hospital of Sichuan University, the Sichuan Cancer Hospital, and the CapitalBio Medlab in Chengdu, China in January 2022. In the discussion, a semi-structured interview guide, whose development was based on a literature review and by consultation with an expert panel, was used by the facilitator of the focus group. The discussion lasted about 60 minutes and a study staff member was present to take notes. After the analysis, which was conducted independently by two research assistants, five attributes were identified for inspection interventions. These attributes were the lab safety inspector, the inspection frequency, the inspection timing, the communication of the outcome, and a follow-up with either a reward or a punishment. The final version of the attributes and levels used in this study can be found in Table 1.

A "lab safety inspector" was defined as a professional, who specialized in evaluating the safety practices of the lab. Having an appropriate safety inspector is crucial in ensuring compliance with relevant safety guidelines to prevent accidents, injuries, and illnesses from occurring. Based on the focus group discussion, four different levels or types of safety inspectors were determined for this attribute. They were the lab director, the group leader, the safety committee member, and the external expert.

The attribute "inspection frequency" referred to how often safety inspections were conducted in a clinical lab. A safety inspection could be conducted regularly or not. A regular safety inspection may help to identify safety hazards and unsafe practices in a timely manner and could provide a consistent level of safety oversight. Based on the discussion of the focus group, four levels were assigned to describe this attribute. They were a weekly inspection, a monthly inspection, an inspection only before an audit, or an inspection after an audit.

The attribute "inspection timing" indicated whether each safety inspection would be scheduled at a fixed (pre-scheduled) or a random time. Unlike a pre-scheduled safety inspection, a randomized inspection would be unpredictable and may help inspectors to see the real safety condition of a lab, but the random occurrence of the inspection may disrupt their operations and undermine a lab's capability for planning.

The attribute "communication of outcome" referred to how a clinical lab provided its workers with feedback about the results of the safety inspections. Feedback, if delivered properly, helps workers to improve their performance. There were various ways of delivering feedback. In this study we considered four feedback delivery options, following the discussion of the focus group. They were either no feedback provided, sending an email to each inspected lab worker, verbally communicating the results with lab workers through their supervisors, or publicly posting the inspection outcomes.

**Table 1. Attributes and levels used in the discrete choice experiment.**

| Attribute | Levels | Variables for Coding | | |
|---|---|---|---|---|
| | | GROUP | SAFETY | EXTERNAL |
| Lab Safety Inspector | Lab director[†] | -1 | -1 | -1 |
| | Group leader | 1 | 0 | 0 |
| | Safety committee member | 0 | 1 | 0 |
| | External expert | 0 | 0 | 1 |
| | | MONTHLY | BEFORE | AFTER |
| Inspection Frequency | Weekly[†] | -1 | -1 | -1 |
| | Monthly | 1 | 0 | 0 |
| | Before an audit | 0 | 1 | 0 |
| | After a safety incident | 0 | 0 | 1 |
| | | RANDOM | | |
| Inspection Timing | Fixed day and time[†] | -1 | | |
| | Random day and time | 1 | | |
| | | EMAIL | SUPERVISOR | POST |
| Communication of Outcome | No feedback on an inspection outcome[†] | -1 | -1 | -1 |
| | Feedback sent by email to each individual | 1 | 0 | 0 |
| | Feedback delivered verbally by a supervisor | 0 | 1 | 0 |
| | Inspection outcome posted publicly | 0 | 0 | 1 |
| | | DISCUSSION | RETRAIN | RECOGNITION |
| Reward / Punishment | No consequence regardless of the inspection outcome[†] | -1 | -1 | -1 |
| | Meet & discuss with a supervisor for an unsatisfactory outcome | 1 | 0 | 0 |
| | Receive retraining following an unsatisfactory outcome | 0 | 1 | 0 |
| | Receive recognition following a satisfactory outcome | 0 | 0 | 1 |

[†]: Reference level

The last attribute "reward/punishment" indicated the consequences of safety inspections. The consequence could either be a reward or a punishment given to lab workers based on their adherence to safety guidelines. Based on the focus group discussion, four levels or consequence options were identified. They were no consequence at all, having a discussion with a supervisor, receiving retraining, and receiving recognition.

In a DCE, it is important to select a reference level for each attribute. The reference level serves as the point of comparison against which all other levels of the attribute are evaluated. It establishes a benchmark for making meaningful comparisons. One commonly used approach to determine the reference level is by opting for the most fundamental level of the attribute [21]. In this study, after careful deliberation with an expert panel, the reference levels for each attribute were determined and presented in Table 1.

In our study, we utilized effects coding. This approach involved creating a number of new variables, which was the number of levels of an attribute being coded, minus one. For instance, the attribute of lab safety inspector had four levels, so we created three new variables, namely GROUP, SAFETY, and EXTERNAL. Since the level "lab director" was defined as the reference level, we set the values of GROUP, SAFETY, and EXTERNAL for "lab director" to be -1, as shown in Table 1.

As can be seen in Table 1, five total experimental attributes were identified, four with four levels and one with two levels. This led to a full factorial design with ($4^4$ x 2) 512 profiles, resulting in (512 x 511 / 2) 130,816 combinations of pair-wise choice sets. Given the large number, a

D-efficient design was employed to maintain optimal orthogonality in a fractional design and to reduce the number of profiles. Using macros in SAS, including Mktruns, MktEX, and ChoiceEff, 32 profiles were selected from this pool of 512 profiles to meet the criteria of a D-efficient design. The selected 32 profiles were then divided into two blocks using Mktblock in SAS; each block contained 8 choice sets, each of which consisted of two profiles. As a result, there were two versions of the DCE questionnaire. Regardless of the version, each questionnaire included demographic questions about the respondents.

The original questionnaire was presented to 2 laboratory directors and 4 workers. The presentation of the choice sets was slightly modified to enhance clarity, after their feedback was received. In addition, an explanation of the choice experiments was given at the beginning of the questionnaire, followed by a demonstration to the participants on how to answer the choice questions.

Ethics approval was granted by the Human Research Ethics Committee of the University of Electronic Science and Technology of China (Approval Number: 1061420211102001) in November 2021. Additional information regarding the ethical, cultural, and scientific considerations specific to inclusivity in global research is included in the S1 File.

## Data collection

Data were collected in 35 clinical laboratories from 7 healthcare institutes in the city of Chengdu, China, from March to July of 2022. During this period, laboratories in Chengdu had returned to normal operation, despite the COVID pandemic. According to the literature, 20–30 respondents per version could provide a precise estimation for the parameters [22]. Since there were two versions of the questionnaire, the appropriate sample size was 40–60. Considering the response rate, 222 laboratory workers were invited.

Participants were randomly selected by using a stratified sampling. The name list was given to each laboratory manager by a study investigator, so that the manager could invite laboratory workers on the list to fill in the questionnaires. To be more specific, the distribution of questionnaires among various healthcare institutes was as follows: the West China Hospital of Sichuan University received 65 questionnaires, the Sichuan Hospital for Women and Children received 36, the Sichuan Cancer Hospital received 26, the No. 3 People's Hospital of Chengdu received 15, the Nuclear Industry No. 416 Hospital received 3, the People's Hospital in Wenjiang District received 23, and the CapitalBio Medlab received 54.

Information about the purpose of the study and the assurance of confidentiality was presented to the participants before they started to fill the questionnaires. To reduce hypothetical bias, defined as the discrepancies between responses identified in the DCE and those identified in reality, the participants were encouraged to answer the questions as if they were making actual decisions. The participation in the study was completely voluntary, and prior to answering the questionnaire, each participant gave their informed consent by signing a written consent form.

## Data analysis

The data were analyzed by Stata V.15.1 for Windows. A conditional logit model was used to investigate the impact of the key attributes of an inspection intervention on the laboratory workers' compliance with safety guidelines. Similar to a cluster analysis, a latent class model (LCM) was used to evaluate the heterogeneity among the respondents. The LCM divided the respondents into several subgroups based on their responses to the choice sets [22] such that respondents in the same subgroup were more similar to each other than they were to respondents in another subgroup. The first step was to determine the appropriate number of

subgroups, which could be determined by assessing a set of performance indicators, such as the Akaike Information Criterion (AIC), the Constrained Akaike Information Criterion (CAIC), and the Bayesian Information Criterion (BIC). The number of subgroups was commonly determined based on the minimum values of these performance indicators [23]. After the number of subgroups was determined, based on the posterior probabilities of membership in each subgroup, each respondent was assigned to the subgroup with the highest probability of membership. In other words, the LCM divided the respondents into subgroups, based on their similar response patterns while also maximizing the differences between these subgroups.

A notable advantage of the DCE is its capability to systematically estimate the willingness of clinical laboratory workers to adhere to safety guidelines in different hypothetical scenarios, all without the need for actual implementation. This estimation can be converted to the probability of individual workers complying with safety guidelines in a specific scenario [24]. This powerful feature enables the identification of best practices that effectively promote safety compliance.

In DCEs, each scenario was analyzed against a *baseline* case, which referred to the scenario where all attributes were set to their reference levels as defined in Table 1. This includes the laboratory safety inspector being the lab director, a weekly inspection frequency with a fixed day and time, no feedback on an inspection outcome, and no consequences irrespective of the inspection outcome.

## Results

### Study population

The questionnaire was completed by 222 respondents with 34 excluded from the analysis due to incomplete or incorrect answers. Data from the remaining 188 (85%) questionnaires were included in the analysis. Table 2 summarizes the socio-demographic details of the participants. Most respondents were female (72.87%), between ages 26 and 35 (54.79%), employed as contractors (71.81%), and worked in a midsized lab with 50–100 people (45.74%).

### Impact of the attributes for a homogeneous population

The coefficients of a conditional logit model are displayed in Table 3. From the table, it can be seen that the laboratory workers had positive responses (when compared with the corresponding references) to having members from the safety committee to conduct the safety inspection (coefficient = 0.2155, p<0.001), receiving monthly inspections (coefficient = 0.2493, p<0.001) and being retrained as the consequence of an unsatisfactory inspection (coefficient = 0.3558, p<0.001). On the other hand, the workers felt less willing to comply with safety guidelines if safety inspections took place only after safety incidents (coefficient = -0.3451, p<0.001). In addition, the workers were insensitive to the feedback of an inspection outcome delivered verbally by a supervisor (coefficient = 0.0914, p>0.1).

### Best practice inspection interventions for a homogeneous population

Implementing an intervention requires a significant investment of time and resources. Furthermore, the outcomes of an intervention are often unpredictable, which increases the risk associated with implementing it. However, owing to the prominent characteristics of DCEs, it becomes possible to predict intervention outcomes through probability analyses. This analytical approach not only enables the anticipation of intervention outcomes, but also aids in identifying the best practice intervention with the greatest probability of success. For example, one

**Table 2. Socio-demographic characteristics of the participants.**

| Characteristics | Percentage of sample (%) |
|---|---|
| Gender | |
| Male | 27.13 |
| Female | 72.87 |
| Age in years | |
| < 26 | 21.81 |
| 26–35 | 54.79 |
| > 36 | 23.40 |
| Education | |
| Junior college or lower | 24.47 |
| Undergraduate | 48.94 |
| Postgraduate or above | 26.59 |
| Years of Working in a Laboratory | |
| < 1 | 18.09 |
| 1–5 | 26.60 |
| 6–10 | 25.53 |
| > 10 | 29.79 |
| Employment Type | |
| Permanent | 17.55 |
| Contract | 71.81 |
| Temporary | 10.64 |
| Size of the Laboratory (No. of people) | |
| < 50 | 26.60 |
| 50–100 | 45.74 |
| > 100 | 27.66 |

**Table 3. Regression results for a conditional logit model.**

| Attributes | Levels | Coefficients | Standard error |
|---|---|---|---|
| Lab Safety Inspector | Group leader | -0.0671 | 0.0630 |
| | Safety Committee member | 0.2155*** | 0.0633 |
| | External expert | -0.1205 | 0.0622 |
| Inspection Frequency | Monthly | 0.2493*** | 0.0644 |
| | Before an audit | -0.0087 | 0.0655 |
| | After a safety incident | -0.3451*** | 0.0620 |
| Inspection Timing | Random day and time | 0.0766* | 0.0312 |
| Communication of Outcome | By an individual email | 0.1776** | 0.0639 |
| | By a supervisor, given verbally | 0.0914 | 0.0599 |
| | Outcome posted publicly | 0.0618 | 0.0604 |
| Reward / Punishment | Meet a supervisor if unsatisfactory | 0.0435 | 0.0633 |
| | Receive retraining if unsatisfactory | 0.3558*** | 0.0655 |
| | Receive recognition if satisfactory | 0.3231*** | 0.0626 |

***$p < 0.001$,

**$p < 0.01$,

*$p < 0.1$

possible intervention could involve using a lab director as the inspector, conducting inspections weekly, at a random day and time, and after the inspection, feedback is sent by email, with no consequence regardless of the inspection outcome; and there are in total 512 (= 4×4×2×4×4) possible combinations of the attribute levels in Table 1. We show in detail how the probability analysis was conducted using the baseline as an example in S3A File. In S3B File, we provide a spreadsheet showing how the probabilities of all 512 intervention scenarios were calculated.

After exhaustively evaluating all possible interventions, the intervention yielding the highest probability of compliance with safety guidelines was identified as the best practice inspection intervention, which included a monthly inspection, but at a random day and time, and conducted by a safety committee member. Furthermore, when the inspection is done, feedback is sent to each laboratory worker by email; if found unsatisfactory, the worker shall receive retraining. The projected compliance of this best practice intervention could improve the probability of compliance from 25.86% at the baseline case (Homogeneous Scenario # 186 in S3B File) to 74.54% (Homogeneous Scenario # 488 in S3B File).

## Impact of the attributes for a heterogeneous population

Additional analyses were conducted by controlling the respondents' demographics, including age, gender, education, years of work experience, employment type, and lab size. The corresponding results were presented in S1–S6 Tables. Each table focuses on a specific demographic variable, dividing the respondents into different groups. Each table shows that the coefficients vary in terms of direction and significance among the different groups, indicating the presence of heterogeneity. To further understand how heterogeneity affected the results across multiple demographics, rather than just one characteristic, a more sophisticated method based on the LCM was employed. Using the LCM, two subgroups (Class A and Class B) of the respondents with distinct responses to safety guidelines were identified, where Class A represented 31.91% of the cohort, and Class B was the remaining 62.09%.

Using Class B as the reference, a regression was conducted to show how the intention of safety compliance for Class A members could be predicted by the demographic characteristics. Table 4 shows the regression results, indicating that the only two characteristics that could significantly differentiate the cohort are whether they were permanently employed and whether they worked in a midsize laboratory with 50–100 workers. Specifically, a permanent employee was more likely to be in Class A (coefficient = 0.9238, $p<0.01$), while a worker in a midsize laboratory was more likely to be in Class B (coefficient = -0.6159, $p<0.01$).

Table 5 shows the regression results within Classes A and B, where each class revealed a distinct intention for compliance. The laboratory workers in Class A showed no significant preference for inspection timing, how the inspection outcome was communicated, or whether or not there was a corresponding reward or punishment. The results in Table 5, however, indicate that Class A members were less likely to comply with safety guidelines if the inspection was carried out by their group leader (coefficient = -0.5874, $p<0.1$); and they were more likely to comply if the inspection was conducted monthly (coefficient = 0.7706, $p<0.01$). Class A members also indicated they were less likely to follow the safety guidelines if they were inspected only after safety incidents (coefficient = -0.9027, $p<0.001$).

On the other hand, for members in Class B, their intention for compliance could be significantly influenced by the inspector, the way in which the outcome was communicated, and by the reward or the punishment. They were more likely to comply if the inspector was a safety committee member (coefficient = 0.3677, $p<0.001$) but were less likely to comply if they were inspected by an external expert (coefficient = -0.4627, $p<0.01$). In terms of the manner of

**Table 4. Two-class membership: Regression results of Class A (31.91%) against the demographic characteristics, with Class B (62.09%) as the reference level.**

| Characteristics | Class A (31.91%) | |
|---|---|---|
| | Coefficient | SE |
| Gender | | |
| Male | Ref | Ref |
| Female | 0.1147 | 0.1993 |
| Age in years | | |
| < 26 | Ref | Ref |
| 26–35 | 0.6000 | 0.5479 |
| > 36 | -0.0568 | 0.3227 |
| Education | | |
| Junior college or lower | Ref | Ref |
| Undergraduate | -0.2392 | 0.3541 |
| Postgraduate or above | -0.0036 | 0.2482 |
| Years of Working in a Laboratory | | |
| < 1 | Ref | Ref |
| 1–5 | -0.5622 | 0.5300 |
| 6–10 | -0.4959 | 0.3598 |
| > 10 | 0.0394 | 0.3354 |
| Employment Type | | |
| Permanent | 0.9238** | 0.4501 |
| Contract | -0.0254 | 0.3518 |
| Temporary | Ref | Ref |
| Size of the Laboratory (No. of people) | | |
| < 50 | Ref | Ref |
| 50–100 | -0.6159** | 0.3096 |
| > 100 | 0.1598 | 0.3333 |

***$p<0.001$,

**$p<0.01$,

*$p<0.1$.

communication, Class B members were more likely to comply if the inspection outcome was conveyed to them verbally by their supervisors (coefficient = 0.3094, $p<0.1$). The result also showed that they were more likely to comply with the safety guidelines if they could receive recognition for a satisfactory outcome (coefficient = 0.5206, $p<0.001$), and retraining if the inspection outcome was unsatisfactory (coefficient = 0.5412, $p<0.001$).

According to the above discussion, clinical lab workers in Class A could be described as more "process-oriented". This is because their preferences were determined mainly by how the inspection was conducted (e.g., the inspector and the frequency of the inspection). On the other hand, Class B lab workers were more "outcome-oriented", as their compliance strongly depended on the consequence of the inspection (e.g., reward or punishment).

## Best practice inspection interventions for a heterogeneous population

Likewise, by exhaustively evaluating all possible interventions, the best practice inspection interventions were identified for laboratory workers in both Classes A and B as the ones that yielded the highest compliance with the safety guidelines. Like the overall best practice

**Table 5. Regression result for a latent class model.**

| Attributes | Levels | Class A | | Class B | |
|---|---|---|---|---|---|
| | | Coefficients | Standard error | Coefficients | Standard error |
| Lab Safety Inspector | By a group leader | -0.5874* | 0.2475 | 0.2162 | 0.1289 |
| | By a safety committee member | -0.0114 | 0.1999 | 0.3677*** | 0.0963 |
| | By an external expert | 0.4243 | 0.2548 | -0.4627** | 0.1461 |
| Inspection Frequency | Monthly | 0.7706** | 0.2667 | 0.0769 | 0.1115 |
| | Before an audit | -0.0670 | 0.1836 | -0.0915 | 0.1259 |
| | After a safety incident | -0.9027*** | 0.2158 | -0.1655 | 0.1273 |
| Inspection Timing | Random day and time | 0.0329 | 0.0837 | 0.0781 | 0.0544 |
| Communication of Outcome | By an individual email | 0.0467 | 0.1553 | 0.2214 | 0.1133 |
| | By a supervisor, given verbally | -0.1860 | 0.1972 | 0.3094* | 0.1213 |
| | Outcome posted publicly | 0.4339 | 0.2316 | -0.0872 | 0.0993 |
| Reward / Punishment | Meet a supervisor if unsatisfactory | 0.1115 | 0.1855 | 0.0489 | 0.1148 |
| | Receive retraining if unsatisfactory | 0.0173 | 0.1750 | 0.5412*** | 0.1364 |
| | Receive recognition if satisfactory | 0.1084 | 0.1593 | 0.5206*** | 0.1412 |

***$p<0.001$,

**$p<0.01$,

*$p<0.1$

intervention, a best practice inspection for Class A included monthly and random inspections. However, the best practice intervention specifically for Class A laboratory workers also included safety inspections to be conducted by an external expert, feedback to be posted publicly, and unsatisfactory performance to be followed up with a discussion with their supervisor. For Class A, this inspection configuration could yield compliance at 85.48% (Class_A Scenario # 301 in S3B File).

As for Class B laboratory workers, they performed best when the inspector was a safety committee member, inspections occurred randomly and weekly, supervisors communicated inspection outcomes verbally, and poor outcomes were responded to with retraining. With this intervention configuration, Class B compliance could be as high as 81.40% (Class_B Scenario # 500 in S3B File). This implementation may be viewed as a best practice for the Class B cohort.

## Discussion

### Principal findings

This study found that a set of five key attributes, including the inspector, the inspection frequency, the inspection timing, the communication of the inspection outcome, and a follow-up with either a reward or a punishment, were critical to the effectiveness of an inspection intervention. In general, retraining and inspection after a safety incident could yield the highest increase and decrease on the compliance from the baseline case, respectively. More specifically, if one could only change a single attribute level from the baseline case, providing retraining when a worker violated safety guidelines could enhance the compliance probability the most from 25.86% to 50.62% (Homogeneous Scenarios # 186 and 188 in S3B File), while conducting inspections only after safety incidents could decrease compliance to the largest extent from 25.86% to 18.20% (Homogeneous Scenarios # 186 and 138 in S3B File). In addition, it was also found that heterogeneous responses from laboratory workers toward an inspection

intervention occurred when the heterogeneity could be characterized by the type of employment and the size of the laboratory. Tailored intervention design would be much more effective than a one-size-fits-all inspection intervention in terms of improving the laboratory workers' intention of compliance with safety guidelines.

## Comparison with other studies and possible explanations

Although an inspection plays a vital role in ensuring workplace safety, a systematic review conducted by Cornish et al. [10] found that the lack of inspection interventions to improve the laboratory workers' compliance was a typical gap in a clinical laboratory's safety management. Our study filled this gap by identifying evidence-based best practice inspection interventions. Our findings indicated that with an appropriate design of an inspection intervention, laboratory workers would be more likely to comply with safety guidelines. This result was consistent with those found in other industries [11–13].

To design an appropriate inspection intervention, our study confirmed that the five attributes considered in this study were crucial to the success of an intervention. The first attribute was about the inspector. As suggested by Woodcock [25], a suitable inspector should be experienced and knowledgeable, so as to have a better understanding about the safety regulations and standards. Additionally, a suitable inspector should be characterized by objectivity to provide unbiased and objective evaluations of compliance. From the perspective of the clinical lab workers, our study supported Woodcock's argument by showing that the person responsible for inspection could affect a worker's willingness to comply with the safety guidelines. Therefore, selecting a suitable inspector is critical to ensure the effectiveness of inspection intervention.

In terms of the inspection frequency, we found that clinical lab workers preferred regular inspections, such as monthly inspections. This might be because regular inspections could translate an organization's commitment to maintain a safe working environment into perceivable actions, and promote a culture of safety in the workplace [26, 27]. Regarding the timing of a safety inspection, although both the fixed and the randomized timings of an inspection had their merits, we found that lab workers preferred random inspections over fixed ones. This finding was consistent with the recommendations made by Berte [28] and Wyllie et al. [29]. One possible reason for this preference was that a randomized inspection was more effective in enhancing the lab workers' safety awareness to motivate them to remain vigilant at all times, thereby reducing the risk of safety incidents. The result that the lab workers preferred regular but randomized safety inspections (e.g., when an inspection occurred on an unspecified day per month) implied that they welcomed random but not too random inspections. This indicates that the workers appreciated a balance between safety practices and smooth business operations.

Finally, providing performance feedback was also a critical component in the design of an inspection intervention. Consistent with previous studies [30, 31], our study found that lab workers were more likely to comply with safety guidelines if they were informed about the inspection results. This is reasonable because having performance feedback is helpful to create a sense of accountability, reinforce safe habits and encourage continued engagement in the safety process. In addition to providing performance feedback, consequences based on performance should be considered as well. We found that compared to no consequences, providing appropriate rewards or punishments could encourage lab workers to comply with safety guidelines. If there were a lack of consequences after a safety inspection, a negative safety culture might arise, where safety was not seen as a priority, leading to poor or no safety awareness among individuals.

Furthermore, in our study, it was found that the laboratory workers in Class A were much more compliant than Class B workers, e.g., 45.23% (Class_A Scenario # 186 in S3B File) vs. 17.17% (Class_B Scenario # 186 in S3B File) in the baseline case. In previous studies, age was used as a predictor for the workers' compliance with safety guidelines [32]. However, our study did not support this conclusion. Rather than age, the employment type and the size of the laboratory were the predictors in our study. We found that Class A members were more likely to be permanent employees and were more likely to be part of a small laboratory environment. Laboratory workers with a permanent work contract could have a stronger sense of belonging and a higher level of organizational commitment, which yielded a higher level of compliance with safety guidelines and regulations. In addition, individuals in a smaller-sized laboratory were less likely to "hide in a crowd" than those in a larger-sized laboratory, which increased task visibility and, therefore, resulted in a higher compliance.

In this study, we found that the responses from the laboratory workers in Class A and Class B were different. For instance, Class A workers preferred inspections by external experts instead of their group leader, indicating that they favored professionalism. Furthermore, Class A workers were less receptive to training and feedback, showing that they may have had more confidence in their professional abilities than Class B workers, who were more likely to have preferred retraining and verbal communication from supervisors. This may indicate that more experienced laboratory workers or those in higher positions are less adaptive to interventions. Laboratory workers come from many different backgrounds; thus, personalization of professional practices can be beneficial. With the inspection customized to their preferences, both classes could achieve high compliance rates (85.48% for Class A, 81.40% for Class B).

## Strengths and limitations

The main strength of this paper is to propose an evidence-based approach to identify best practice inspection interventions while considering the perspective of laboratory workers, which can be extremely useful when a healthcare facility is contemplating the introduction of new interventions. Since an intervention is implemented to ensure the laboratory workers' compliance, their preferences and responses to the intervention ought to be considered. As shown in this paper, the workers' responses to interventions may be heterogeneous, which makes it difficult to predict the outcomes of new interventions. Uncertain outcomes may create additional stress and pressure for the workers. A failed trial may increase the difficulty for the workers to seriously adapt to future changes. The proposed DCE presents a new "*intervention-without-implementation*" approach that can be used to mitigate the risks associated with implementing new interventions. Using this approach, decision-makers can obtain ex-ante insights about the effects of newly introduced interventions and observe how the effects vary according to changing factors and individual characteristics. A probability analysis can be applied to predict compliance, given an intervention's design, which helps to identify best practices that yield satisfactory compliance.

There are several limitations in our study. First, the laboratory workers' responses to inspection interventions were examined in hypothetical scenarios. It would be interesting to measure and compare their responses in actual life. That said, this study provides valuable preliminary understanding of the laboratory workers' responses to five key intervention attributes. There may be other attributes that can also affect the laboratory workers' compliance that were not considered in this study. In addition, this study does not provide a full answer about how the key attributes of the interventions enhance the laboratory workers' compliance. This approach does not guarantee that after a high compliance rate is reached, it will be sustainable.

### Implications to clinicians and administrators

Our findings indicate the importance of an employee-centered approach, rather than a top-down strategy, to reduce the risk of introducing new interventions. The employee-centered approach is also helpful in designing customized interventions by providing information about the individuals' heterogeneous responses to the interventions. In recent studies, customization has been recognized as a promoting factor for intervention success, and, therefore, has started to receive attention in intervention design [33, 34].

In practice, customization could be conducted at both the laboratory and individual levels. Take the intervention of using safety committee members as lab safety inspectors as an example. If a clinical laboratory predominantly comprised individuals from Class A, who were relatively insensitive to this intervention, implementing this intervention might not significantly improve compliance. Conversely, in a clinical laboratory with most individuals from Class B, who favored being inspected by safety committee members, this intervention could be considered appropriate. In situations where the proportion of laboratory workers in each class was evenly distributed, the laboratory could choose interventions that were well-received by both classes to enhance overall compliance.

Furthermore, customization of interventions can also be implemented at the individual level. For example, the intervention involving receiving recognition for satisfactory inspection outcomes received heterogeneous responses. As a result, the laboratory should consider requiring only those individuals who had positive attitudes towards this intervention to sign up for the inspection outcome.

This study also implies that the laboratory workers recognized important factors, such as retraining and giving recognition as a means to make them more compliant with the biosafety guidelines. Even for both Classes A and B, the LCM showed a heightened probability of compliance to such scenarios, despite their perceived differences. In terms of retraining, the nature of the work in a clinical laboratory may be evolving and fast changing, therefore, requiring the laboratory workers to learn, unlearn, and relearn to keep up-to-date [35]. This study was conducted during a period when COVID had subsided in China and laboratories had returned to operating normally. The subjects of this study had just had a renewed experience on the importance of retraining. As indicated by Marin [36], in the time of the COVID pandemic, no one could be exempt from unlearning old habits, because relearning is the new basis of how we work, and adapt to new environments. Embracing these practices can influence not only the acceptance of interventions but also increase open-mindedness to future possibilities. Similarly, recognition of performance can potentially improve job satisfaction for lab personnel and set examples for other members while improving intervention success.

### Conclusion

This study considers an intervention design for lab safety inspections. Five key intervention attributes were identified. It was demonstrated that each attribute had a significant impact on a laboratory worker's behavior toward compliance. Using latent class analysis, the laboratory workers were classified into two classes with distinct compliant behaviors and responses in the intervention. A unique best practice intervention that yielded the highest predicted compliance was identified for each class. Although, initially, the workers in one class were much more compliant than those in the other class in the baseline case, both best practice interventions resulted in satisfactory compliance rates. The conclusion of this study is that it is critical for decision-makers to understand a worker's responses for intervention attributes in order to develop a suitable intervention that can induce high compliance.

## Supporting information

**S1 File. Inclusivity in global research.**
(DOCX)

**S2 File. Complete survey data of 188 laboratory workers.**
(XLSX)

**S3 File.** A. Probability calculation of a scenario–An example. B. Probability calculation for all scenarios.
(ZIP)

**S1 Table. Regression results for various groups based on age.**
(DOCX)

**S2 Table. Regression results for various groups based on gender.**
(DOCX)

**S3 Table. Regression results for various groups based on education.**
(DOCX)

**S4 Table. Regression results for various groups based on years of work experience.**
(DOCX)

**S5 Table. Regression results for various groups based on employment type.**
(DOCX)

**S6 Table. Regression results for various groups based on lab size.**
(DOCX)

**S1 Questionnaire.**
(DOCX)

**S2 Questionnaire.**
(DOCX)

## Acknowledgments

The authors would like to acknowledge and thank all clinical laboratory workers who participated in this study.

## Author Contributions

**Conceptualization:** Qiaoling Qin, Cynthia Tseng, Wenlin Chen, Chung-Li Tseng.

**Data curation:** Qiaoling Qin.

**Formal analysis:** Wenlin Chen.

**Investigation:** Qiaoling Qin, Cynthia Tseng, Chung-Li Tseng.

**Methodology:** Qiaoling Qin.

**Resources:** Qiaoling Qin.

**Supervision:** Chung-Li Tseng.

**Validation:** Cynthia Tseng.

**Visualization:** Cynthia Tseng.

**Writing – original draft:** Qiaoling Qin, Cynthia Tseng, Wenlin Chen, Chung-Li Tseng.

**Writing – review & editing:** Wenlin Chen, Chung-Li Tseng.

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
