## [Decision Letter · Decision Letter 0]

3 Apr 2023

PONE-D-22-26396Best Practices for Implementing Biosafety Inspections in a Clinical Laboratory: Evidence from a Multi-Site Experimental StudyPLOS ONE

Dear Dr. Chen,

Thank you for submitting your manuscript to PLOS ONE. After careful consideration, we feel that it has merit but does not fully meet PLOS ONE’s publication criteria as it currently stands. Therefore, we invite you to submit a revised version of the manuscript that addresses the points raised during the review process.

We look forward to receiving your revised manuscript.

Kind regards,

Bruna Moreno Dias, PhD

Academic Editor

PLOS ONE

Journal Requirements:

3. Please include your ethics statement in the Methods section of your manuscript. In the Methods section of your revised manuscript, please include the full name of the institutional review board or ethics committee that approved the protocol, the approval or permit number that was issued, and the date that approval was granted.

4. Please include a complete copy of PLOS’ questionnaire on inclusivity in global research in your revised manuscript. Our policy for research in this area aims to improve transparency in the reporting of research performed outside of researchers’ own country or community. The policy applies to researchers who have travelled to a different country to conduct research, research with Indigenous populations or their lands, and research on cultural artefacts. The questionnaire can also be requested at the journal’s discretion for any other submissions, even if these conditions are not met.  Please find more information on the policy and a link to download a blank copy of the questionnaire here: https://journals.plos.org/plosone/s/best-practices-in-research-reporting. Please upload a completed version of your questionnaire as Supporting Information when you resubmit your manuscript.

This work was supported in part by a grant from the National Natural Science Foundation of China (grant number: 71902017).

Additional Editor Comments:

I understand that the manuscript is interesting and pertinent. However, adjustments are needed to better understand the study.

Please consider the reviewers' comments carefully.

I further suggest that:

In the abstract, the method needs to be better described.

Methods need to be improved, especially about attributes and levels and about A and B classes.

In the results, I believe that other data and results could be incorporated for a better understanding of the presented data of compliance.

The discussion can be broadened, with more foundation in the literature.

Reviewers' comments:

Reviewer's Responses to Questions

**Comments to the Author**

1. Is the manuscript technically sound, and do the data support the conclusions?

Reviewer #1: Partly

Reviewer #2: Yes

2. Has the statistical analysis been performed appropriately and rigorously? 

Reviewer #1: Yes

Reviewer #2: I Don't Know

3. Have the authors made all data underlying the findings in their manuscript fully available?

Reviewer #1: Yes

Reviewer #2: No

4. Is the manuscript presented in an intelligible fashion and written in standard English?

Reviewer #1: Yes

Reviewer #2: No

5. Review Comments to the Author

Reviewer #1: PONE-D-22-26396

Best Practices for Implementing Biosafety Inspections in a Clinical Laboratory:

Evidence from a Multi-Site Experimental Study

In this paper, the authors contribute to the existing literature by investigating the role of different inspection intervention attributes on laboratory safety compliance using the DCE of 188 clinic workers in the city of Chengdu (China). These attributes include the inspector, the inspection frequency, the inspection timing, the communication of the inspection outcome and either a reward or punishment. One interesting finding is that providing retraining when workers violated safety guidelines (in reward/punishment attributes) yields the highest increase in compliance probability.

Major points

I think the authors succeeded in the approach. However, there are some major points that I would like to ask the authors to further clarify or improve:

1. Introduction:

a. The authors should clearly emphasize their contributions to the existing literature in the introduction rather than in the discussions.

2. Method:

a. It should be an additional subsection where the authors discuss the attributes and their levels.

b. The authors should explain in more detail how the baseline scenario was defined. Since the authors mentioned that

“To achieve this goal, the baseline case as the scenario in which all the attributes were set to their reference levels (see Table 1) was defined”.

So, did the author mention that the baseline/status quo option is where all attributes were at the reference level?

However, I observed that in the questionnaire (Supplementary material) the authors said that there were only two alternative scenarios A and B that the respondents can choose. Is there no status quo option in each choice question that respondents can choose in case they prefer neither A nor B?

c. Following this, how the baseline of the dependent variable in the Probability model can be defined? Is Scenario A defined as a baseline? The authors should carefully verify this issue; otherwise, the results are problematic.

3. Results and discussions:

a. Does Table 3 present the results without controlling for individuals’ demographic characteristics? If yes, it will be beneficial to have an estimation result with the control variables.

b. It is better to specify the difference between Classes A and B. For instance, can we say that respondents in Class B are more “reward/punishment-sensitive” than those in Class A? and that Class A is a “compliant group”?

c. The authors argued that class A laboratory workers preferred for safety inspections to be conducted by an external expert. However, the Results of Table 5 suggest only a negative and significant impact of “By a group leader” at a 10% level. So, they should be indifferent to “Lab safety inspectors” at a 5% significant level.

Minors

1. The authors should explain what is meaning of the coding matrix (-1,1,0) in Table 1.

2. The first paragraph of the Strengths of the paper is better to be discussed in the introduction.

Reviewer #2: As mentioned by the authors of this manuscript, the biosafety inspection is one of the most important sections in biosafety management. However, the compliance with the guidelines has not been achieved well in many clinical laboratories or biosafety laboratories. This manuscript offered an approach to identify practice inspection interventions, which could be used to identify and assess attributes to improve biosafety management further. Meanwhile, the authors also made some important evidence-based points for intervention design, such as tailored but not one-size-fits-all inspection intervention, an employee-centered approach, rather than a top-down strategy, to reduce the risk of introducing new interventions. All of these would be beneficial to laboratory worker’s compliance with the biosafety management.

There are some comments that might be addressed in a revision:

1. Please spelled out the full name, such as HCW (Line 86), SE (Table 3-5).

2. About Class A and Class B: Please describe in detail the Class A and Class B, including how to divide the Class A and Class B in Method section, the characteristics of Class A and Class B. It is necessary to make these clear for reader.

3. Results description is not very clear. For example, the baseline case, and the calculation the percentage (Line 189-191, Line 235-236, Line 240-242, Line 250-254, Line 269-270…).

4. The English is not of sufficient quality for an international peer-reviewed journal.

6. PLOS authors have the option to publish the peer review history of their article (what does this mean?). If published, this will include your full peer review and any attached files.

Reviewer #1: No

Reviewer #2: No

---

## [Author Response · Author response to Decision Letter 0]

19 May 2023

The authors would like to thank the review team for their thoughtful, thorough, and constructive feedback on our previous submission. We appreciate the opportunity to further strengthen our research. We have taken all reviewer comments to heart and tried our best to address all the raised concerns.

Responses to the Academic editor

>>> Thank you for bringing this to our attention. We have carefully reviewed this manuscript to ensure that it adheres to PLOS ONE’s style requirements. 

2. Please provide additional details regarding participant consent. In the ethics statement in the Methods and online submission information, please ensure that you have specified (1) whether consent was informed and (2) what type you obtained (for instance, written or verbal, and if verbal, how it was documented and witnessed). If your study included minors, state whether you obtained consent from parents or guardians. If the need for consent was waived by the ethics committee, please include this information. If you are reporting a retrospective study of medical records or archived samples, please ensure that you have discussed whether all data were fully anonymized before you accessed them and/or whether the IRB or ethics committee waived the requirement for informed consent. If patients provided informed written consent to have data from their medical records used in research, please include this information.

>>> The individuals who took part in this research were clinical laboratory workers. Their involvement in the study was completely voluntary, and prior to answering the questionnaire, each participant gave their informed consent by signing a written consent form.

The above information has been added to the Methods section of the manuscript on page 7. 

3. Please include your ethics statement in the Methods section of your manuscript. In the Methods section of your revised manuscript, please include the full name of the institutional review board or ethics committee that approved the protocol, the approval or permit number that was issued, and the date that approval was granted.

>>> An ethics approval was granted by the Human Research Ethics Committee of the University of Electronic Science and Technology of China (Approval Number: 1061420211102001) in November 2021.

The above information has been added to the Methods section of the manuscript on page 7. 

4. Please include a complete copy of PLOS’ questionnaire on inclusivity in global research in your revised manuscript. Our policy for research in this area aims to improve transparency in the reporting of research performed outside of researchers’ own country or community. The policy applies to researchers who have travelled to a different country to conduct research, research with Indigenous populations or their lands, and research on cultural artefacts. The questionnaire can also be requested at the journal’s discretion for any other submissions, even if these conditions are not met. Please find more information on the policy and a link to download a blank copy of the questionnaire here: https://journals.plos.org/plosone/s/best-practices-in-research-reporting. Please upload a completed version of your questionnaire as Supporting Information when you resubmit your manuscript.

>>> A complete copy of the questionnaire has been provided as supporting information S1.

This work was supported in part by a grant from the National Natural Science Foundation of China (grant number: 71902017). Please state what role the funders took in the study. If the funders had no role, please state: "The funders had no role in study design, data collection and analysis, decision to publish, or preparation of the manuscript." If this statement is not correct you must amend it as needed. Please include this amended Role of Funder statement in your cover letter; we will change the online submission form on your behalf.

>>> The funders had no role in study design, data collection and analysis, decision to publish, or preparation of the manuscript. This has been added to the financial disclosure on P. 17. 

6. Please include captions for your Supporting Information files at the end of your manuscript, and update any in-text citations to match accordingly. Please see our Supporting Information 

>>> Captions for the Supporting Information files have been provided at the end of the manuscript. 

7. In the abstract, the method needs to be better described.

>>> The Method section in the abstract has been enhanced as follows.

Five key components of an inspection intervention, identified from a focus group discussion, were used as the attributes of a discrete choice experiment (DCE). In the DCE, participants were presented with two hypothetical scenarios and asked to choose the scenario in which they were more willing to comply with the laboratory safety guidelines. Data were collected from 35 clinical laboratories in seven healthcare institutes located in Chengdu, China. In total, 188 laboratory workers completed the DCE. The collected data were analyzed using conditional logit regression and latent class analysis.

8. Methods need to be improved, especially about attributes and levels and about A and B classes.

>>> The Methods section has been revised to include more details. Firstly, additional information has been provided regarding the definitions of each attribute and their corresponding levels in the Discrete Choice Experiment (DCE). This ensures clarity and facilitates a better understanding of the experimental design.

Moreover, the steps involved in utilizing the latent class model (LCM) to divide respondents into class A and class B have been outlined. These steps include specifying the LCM, estimating its parameters, and assigning respondents to their respective subgroups based on their posterior probabilities of membership.

Please refer to the responses provided for Question 2a by Reviewer 1 and Question 2 by Reviewer 2 for more details. 

9. In the results, I believe that other data and results could be incorporated for a better understanding of the presented data of compliance.

>>> In the Results section, we have incorporated the participants' demographic information, including age, gender, education, years of work experience, employment type, and lab size, for further analysis. These results are presented in six tables as part of the Supporting Information. Each table focuses on a specific demographic variable, dividing the respondents into different groups. Each table shows that the coefficients vary in terms of direction and significance among the different groups, indicating the presence of heterogeneity. For more detailed information, please refer to the response to Question 3a from Reviewer 1.

10. The discussion can be broadened, with more foundation in the literature.

>>> Additional discussion regarding the attributes and findings of this study has been included in the Discussion section on page 14, as outlined below.

To design an appropriate inspection intervention, our study confirmed that the three attributes considered in this study were crucial to the success of an intervention. The first attribute was about the inspector. As suggested by Woodcock (24), a suitable inspector should be experienced and knowledgeable, so as to have a better understanding about the safety regulations and standards. Additionally, a suitable inspector should be characterized by objectivity to provide unbiased and objective evaluations of compliance. From the perspective of the clinical lab workers, our study supported Woodcock’s argument by showing that the person responsible for inspection could affect a worker’s willingness to comply with the safety guidelines. Therefore, selecting a suitable inspector is critical to ensure the effectiveness of inspection intervention.

In terms of the inspection frequency, we found that clinical lab workers preferred regular inspections, such as monthly inspections. This might be because regular inspections could translate an organization’s commitment to maintain a safe working environment into perceivable actions, and promote a culture of safety in the workplace (25, 26). Regarding the timing of a safety inspection, although both the fixed and the randomized timings of an inspection had their merits, we found that lab workers preferred random inspections over fixed ones. This finding was consistent with the recommendations made by Berte (27) and Wyllie et al. (28). One possible reason for this preference was that a randomized inspection was more effective in enhancing the lab workers’ safety awareness to motivate them to remain vigilant at all times, thereby reducing the risk of safety incidents. The result that the lab workers preferred regular but randomized safety inspections (e.g., when an inspection occurred on an unspecified day per month) implied that they welcomed random but not too random inspections. This indicates that the workers appreciated a balance between safety practices and smooth business operations.

Finally, providing performance feedback was also a critical component in the design of an inspection intervention. Consistent with previous studies (29, 30), our study found that lab workers were more likely to comply with safety guidelines if they were informed about the inspection results. This is reasonable because having performance feedback is helpful to create a sense of accountability, reinforce safe habits and encourage continued engagement in the safety process. In addition to providing performance feedback, consequences based on performance should be considered as well. We found that compared to no consequences, providing appropriate rewards or punishments could encourage lab workers to comply with safety guidelines. If there were a lack of consequences after a safety inspection, a negative safety culture might arise, where safety was not seen as a priority, leading to poor or no safety awareness among individuals.

 

Responses to Reviewer 1

Major areas of potential improvement

1. Introduction:

a. The authors should clearly emphasize their contributions to the existing literature in the introduction rather than in the discussions.

>>> To emphasize our contributions to the existing literature, we wrote the following paragraph in the Introduction on page 3 and page 4.

Our study contributes to the extant literature in several ways. First, we addressed a significant gap in the safety management of clinical laboratories by identifying evidence-based best practices for inspection interventions. Secondly, we adopted an employee-centered approach, rather than a top-down approach, to gather solid evidence for identifying the most suitable interventions. Specifically, these inspection interventions systematically addressed the clinical lab workers’ preferences to the interventions. Since the ultimate goal of an intervention is to ensure that lab workers are more willing to comply with biosafety guidelines, understanding how clinical lab workers respond to interventions is essential and indispensable when designing effective interventions.

2. Method:

a. It should be an additional subsection where the authors discuss the attributes and their levels.

>>> The following paragraphs about the definitions of each attribute and their corresponding levels have been added to the Methods section on page 5 and page 6.

A “lab safety inspector” was defined as a professional, who specialized in evaluating the safety practices of the lab. Having an appropriate safety inspector is crucial in ensuring compliance with relevant safety guidelines to prevent accidents, injuries, and illnesses from occurring. Based on the focus group discussion, four different levels or types of safety inspectors were determined for this attribute. They were the lab director, the group leader, the safety committee member, and the external expert. 

The attribute “inspection frequency” referred to how often safety inspections were conducted in a clinical lab. A safety inspection could be conducted regularly or not. A regular safety inspection may help to identify safety hazards and unsafe practices in a timely manner and could provide a consistent level of safety oversight. Based on the discussion of the focus group, four levels were assigned to describe this attribute. They were a weekly inspection, a monthly inspection, an inspection only before an audit, or an inspection after an audit.

The attribute “inspection timing” indicated whether each safety inspection would be scheduled at a fixed (pre-scheduled) or a random time. Unlike a pre-scheduled safety inspection, a randomized inspection would be unpredictable and may help inspectors to see the real safety condition of a lab, but the random occurrence of the inspection may disrupt their operations and undermine a lab’s capability for planning. 

The attribute “communication of outcome” referred to how a clinical lab provided its workers with feedback about the results of the safety inspections. Feedback, if delivered properly, helps workers to improve their performance. There were various ways of delivering feedback. In this study we considered four feedback delivery options, following the discussion of the focus group. They were either no feedback provided, sending an email to each inspected lab worker, verbally communicating the results with lab workers through their supervisors, or publicly posting the inspection outcomes. 

The last attribute “reward/punishment” indicated the consequences of safety inspections. The consequence could either be a reward or a punishment given to lab workers based on their adherence to safety guidelines. Based on the focus group discussion, four levels or consequence options were identified. They were no consequence at all, having a discussion with a supervisor, receiving retraining, and receiving recognition. 

b. The authors should explain in more detail how the baseline scenario was defined. Since the authors mentioned that “To achieve this goal, the baseline case as the scenario in which all the attributes were set to their reference levels (see Table 1) was defined”. So, did the author mention that the baseline/status quo option is where all attributes were at the reference level? However, I observed that in the questionnaire (Supplementary material) the authors said that there were only two alternative scenarios A and B that the respondents can choose. Is there no status quo option in each choice question that respondents can choose in case they prefer neither A nor B?

>>> In this paper, the baseline does not represent the status quo. In a DCE, there is no need to establish a status quo, as participants are given hypothetical scenarios to make their trade-off decisions among different attribute levels. We used the strategy to pick one fundamental level of an attribute as its reference level, which is a common practice in conducting DCEs.

Furthermore, in the discrete choice experiment with binary options (e.g., A or B), where the pairwise comparison is easy to interpret, either A is preferred to B, or vice versa. Even in a situation when the participants preferred neither A nor B, the participants were still asked to select one option, A or B, with which they were more willing to comply. This requirement was clearly mentioned in the guidelines of the questionnaire and was emphasized in the verbal introduction to the participants prior to the survey. Theoretically, the inclusion of this mandatory choice has several advantages. First, it promotes active engagement from the respondents, encouraging them to carefully consider their choices and provide more comprehensive information about their preferences. Secondly, the mandatory choice contributes to noise reduction in the data. Lastly, the mandatory choice facilitates a better understanding of the trade-offs the participants are willing to make. Overall, the inclusion of a mandatory choice enhances the quality and reliability of the data, leading to a more robust analysis of the respondents' preferences.

c. Following this, how the baseline of the dependent variable in the Probability model can be defined? Is Scenario A defined as a baseline? The authors should carefully verify this issue; otherwise, the results are problematic.

>>> As we have clarified that scenario A was not considered as the baseline case in this study in the previous question 2b, here we answer how the probability model was constructed. This explanation has been provided in Supporting Information S2.

3. Results and discussions:

a. Does Table 3 present the results without controlling for individuals’ demographic characteristics? If yes, it will be beneficial to have an estimation result with the control variables.

>>> Yes. The results presented in Table 3 did not control the individuals’ demographics. But we have now provided additional analyses by incorporating the participants’ demographic information as control variables. The additional results are presented in Tables S3 – S8. Using Table S3 as an example, which presented the regression results with respect to the respondents' age, the coefficients for the three age groups differed in terms of both direction and significance, implying the presence of heterogeneity across the three different age groups. Heterogeneity can also be observed in the other five tables S4– S8 included in the appendix. Overall, in these six tables in the appendix, only one single demographic characteristic was considered each time. To further understand how heterogeneity affected the results across multiple demographics, rather than just one characteristic, a more sophisticated method based on latent class analysis was employed in the paper.

The above information has been provided in the section of Results on page 10. 

b. It is better to specify the difference between Classes A and B. For instance, can we say that respondents in Class B are more “reward/punishment-sensitive” than those in Class A? and that Class A is a “compliant group”?

>>> Thank you for the comment, which is well taken. It’s true that the class B laboratory workers would be more willing to comply with safety guidelines if there were punishments or rewards after the inspection. They were indeed more reward/punishment-sensitive and we described them as more outcome-oriented in the paper. As for class A workers, indeed they would have a high compliance rate of 85.48% under the best practice inspection intervention. But class B could achieve a high compliance rate as well at 81.4% under the corresponding best practice. As a result, we do not label class A as a “compliant group”, but a more “process-oriented” group, as the workers were more concerned about the process of the inspection, such as who conducted the inspection and the frequency of inspection (from Table 5). 

We have elaborated the difference between classes A and B, and have added a new paragraph in the Results section on page 12, which is also documented below. 

According to the above discussion, clinical lab workers in class A could be described as more “process-oriented”. This is because their preferences were determined mainly by how the inspection was conducted (e.g., the inspector and the frequency of the inspection). On the other hand, class B lab workers were more “outcome-oriented”, as their compliance strongly depended on the consequence of the inspection (e.g., reward or punishment).

c. The authors argued that class A laboratory workers preferred for safety inspections to be conducted by an external expert. However, the Results of Table 5 suggest only a negative and significant impact of “By a group leader” at a 10% level. So, they should be indifferent to “Lab safety inspectors” at a 5% significant level.

>>> We agree with the reviewer’s observation that class A lab workers did not display any significant preference for the type of inspector at the 5% significant level. On the other hand, the probability model that makes use of the regression results to project the probability of compliance under each hypothetical setting is valid, as it is a well-established method. It’s also true that the obtained best practice inspection intervention involved using an external expert as the inspector. Therefore, we made the following changes in the manuscript to address your concern: (i) we stated that the best practice inspection interventions were identified to yield the highest projected compliance with the safety guidelines; and (ii) the best practice intervention for class A laboratory workers “involved” using an external expert as the inspector. We no longer mention that class A workers “preferred” using an external expert as the inspector. We hope that you will agree with our revision. 

Minor areas of potential improvement

1. The authors should explain what is meaning of the coding matrix (-1,1,0) in Table 1.

>>> Table 1 has been revised to clarify the meaning of the coding matrix. An additional paragraph has been added in the Methods section on page 6 for clarification. 

In our study, we utilized effects coding. This approach involved creating a number of new variables, which was the number of the levels of an attribute being coded, minus one. For instance, the attribute of lab safety inspector had four levels, so we created three new variables, namely GROUP, SAFETY, and EXTERNAL. Since the level “lab director” was defined as the reference level, we set the values of GROUP, SAFETY, and EXTERNAL for “lab director” to be -1, as shown in Table 1.

2. The first paragraph of the Strengths of the paper is better to be discussed in the introduction.

>>> As suggested by the reviewer, the first paragraph of the Strengths of the paper has been moved to the introduction. 

 

Responses to Reviewer 2

1. Please spelled out the full name, such as HCW (Line 86), SE (Table 3-5).

>>> The full name with abbreviation has been provided. For instance, HCW stands for health care worker, and SE is an abbreviation for standard error. 

2. About Class A and Class B: Please describe in detail Class A and Class B, including how to divide the Class A and Class B in Method section, the characteristics of Class A and Class B. It is necessary to make these clear for reader.

>>> The following paragraph, which explains how to divide Class A and Class B, has been improved in the Method section on page 7 and page 8. 

The data were analyzed by Stata V.15.1 for Windows. A conditional logit model was used to investigate the impact of the key attributes of an inspection intervention on the laboratory workers’ compliance with safety guidelines. In addition, a latent class model (LCM) was used to evaluate the heterogeneity among the respondents. The LCM divided the respondents into several subgroups based on their responses to the choice sets (22). The appropriate number of subgroups was determined by assessing a set of performance indicators, such as the Akaike Information Criterion (AIC), the Constrained Akaike Information Criterion (CAIC), and the Bayesian Information Criterion (BIC). The number of segments with the minimum values of performance indicators was preferred (23). After the number of subgroups was determined, based on the posterior probabilities of membership in each subgroup, each respondent was assigned to the subgroup with the highest probability of membership. In other words, the LCM divided the respondents into subgroups based on their similar response patterns while also maximizing the differences between these subgroups.

3. Results description is not very clear. For example, the baseline case, and the calculation the percentage (Line 189-191, Line 235-236, Line 240-242, Line 250-254, Line 269-270…).

>>> The explanations for the baseline case (Line 189-191 in the previous version) and the calculation of the probability of a hypothetical intervention scenario (Line 235-236, Line 240-242, Line 250-254, Line 269-270 in the previous version) have been addressed in the revision. We have added a paragraph at the end of the data analysis section on page 8 to briefly describe how the Discrete Choice Experiment (DCE) methodology can be used to project the compliance (or probability) of a hypothetical intervention scenario. We have also provided how the probability of a scenario is calculated with an example in the Supporting Information S2. On page 10, we now describe how we evaluated all 512 possible intervention scenarios to identify the best practice, which is applied to both the homogeneous and classified cases. 

4. The English is not of sufficient quality for an international peer-reviewed journal.

>>> The revision has been reviewed and edited by a professional editor, who is a native English speaker.

---

## [Decision Letter · Decision Letter 1]

19 Jul 2023

PONE-D-22-26396R1Best Practices for Implementing Biosafety Inspections in a Clinical Laboratory: Evidence from a Multi-Site Experimental StudyPLOS ONE

Dear Dr. Chen,

Thank you for submitting your manuscript to PLOS ONE. After careful consideration, we feel that it has merit but does not fully meet PLOS ONE’s publication criteria as it currently stands. Therefore, we invite you to submit a revised version of the manuscript that addresses the points raised during the review process.

 In addition to addressing the reviewer's comments, it is essential to ensure a clear and comprehensive explanation of data availability within the content. Furthermore, provide a detailed justification for the choice of appropriate statistical analysis used, demonstrating its accuracy and validity.

 Please submit your revised manuscript by Sep 02 2023 11:59PM. If you will need more time than this to complete your revisions, please reply to this message or contact the journal office at plosone@plos.org. Please include the following items when submitting your revised manuscript:A rebuttal letter that responds to each point raised by the academic editor and reviewer(s). You should upload this letter as a separate file labeled 'Response to Reviewers'.A marked-up copy of your manuscript that highlights changes made to the original version. You should upload this as a separate file labeled 'Revised Manuscript with Track Changes'.An unmarked version of your revised paper without tracked changes. You should upload this as a separate file labeled 'Manuscript'.If applicable, we recommend that you deposit your laboratory protocols in protocols.io to enhance the reproducibility of your results. Protocols.io assigns your protocol its own identifier (DOI) so that it can be cited independently in the future. For instructions see: https://journals.plos.org/plosone/s/submission-guidelines#loc-laboratory-protocols. Additionally, PLOS ONE offers an option for publishing peer-reviewed Lab Protocol articles, which describe protocols hosted on protocols.io. Read more information on sharing protocols at https://plos.org/protocols?utm_medium=editorial-email&utm_source=authorletters&utm_campaign=protocols.

We look forward to receiving your revised manuscript.

Kind regards,

Ehsan Kamrani

Academic Editor

PLOS ONE

Journal Requirements:

Additional Editor Comments :

Please ensure the clear and comprehensive explanation of data availability in the content and provide a detailed justification on the appropriate statistical analysis employed, demonstrating its correctness and validity.

Reviewers' comments:

Reviewer's Responses to Questions

**Comments to the Author**

1. If the authors have adequately addressed your comments raised in a previous round of review and you feel that this manuscript is now acceptable for publication, you may indicate that here to bypass the “Comments to the Author” section, enter your conflict of interest statement in the “Confidential to Editor” section, and submit your "Accept" recommendation.

Reviewer #1: All comments have been addressed

Reviewer #2: (No Response)

2. Is the manuscript technically sound, and do the data support the conclusions?

Reviewer #1: Yes

Reviewer #2: Yes

3. Has the statistical analysis been performed appropriately and rigorously? 

Reviewer #1: Yes

Reviewer #2: I Don't Know

4. Have the authors made all data underlying the findings in their manuscript fully available?

Reviewer #1: No

Reviewer #2: No

5. Is the manuscript presented in an intelligible fashion and written in standard English?

Reviewer #1: Yes

Reviewer #2: Yes

6. Review Comments to the Author

Reviewer #1: Thank you for the authors' responses. I believe that the revised manuscript at this stage is suitable for publication.

Reviewer #2: This study provides valuable approach for designing best practice inspection interventions, and it these interventions would play important role for biosafety management of clinical laboratories and biosafety laboratories. It would improve the compliance with biosafety guidelines through identifying evidence-based suitable interventions, rather than current top-down approach mainly used. There is a need to descript the tailed methods and related results for this approach in the manuscript for wider application in actual life.

Minor revision (revised manuscript with track changes):

1. Line 48: “to aid physicians in in the diagnosis…”, please delete the extra “in”.

2. About Class A and Class B: The description for them remains unclear. It may be easier to understand in Method section, such as “In other words, the LCM divided the respondents into subgroups, such as Class A and Class B, based on their similar response patterns while also maximizing the differences between these subgroups.” (Line 224-226). Or these subgroups can be tried to explain in Results section: According to the LCM, two subgroups, Class A and B, were identified based on their similar characteristics related to their safety guidelines compliance intentions, where …… (Line 295-297). Please consider seriously and make an appropriate revision.

3. Some important results could be descripted more clearly like S2, such as “For class A, this inspection configuration could yield compliance at 85.48%. (Line 343-344)”,“With this intervention configuration, class B compliance could be as high as 81.40%. (Line 350-351)”, Line 360-364, Line 414-415, Line 281-283 etc. After all, there is a need to explain clearly to a wider audience how these data were obtained, and it may be better to present these results in a table rather than just words.

4. The author mentions the “baseline” several times in the manuscript, but it remains unclear what it is. It is necessary to explain it or give an appropriate description.

7. PLOS authors have the option to publish the peer review history of their article (what does this mean?). If published, this will include your full peer review and any attached files.

Reviewer #1: **Yes: **TIET TONG TUYEN

Reviewer #2: No

---

## [Author Response · Author response to Decision Letter 1]

3 Sep 2023

Responses to the Reviewers’ Comments

The authors would like to thank the review team for their thoughtful, thorough, and constructive feedback on our previous submission. We appreciate the opportunity to further strengthen our research. 

Responses to the Academic editor

>>> Thank you for the feedback. We have carefully reviewed our reference list and ensured that it is complete and correct. We can confirm that none of the cited papers in our manuscript have been retracted and all references are relevant and current. 

2. Please ensure the clear and comprehensive explanation of data availability in the content and provide a detailed justification on the appropriate statistical analysis employed, demonstrating its correctness and validity.

>>> Thank you for your feedback. We have addressed the issues of data availability and the appropriate statistical analysis in the revised manuscript. Regarding data availability, we have now provided all data, including the survey data in Supporting Information S2, and detailed evaluation of all 512 scenarios for determining best practices in Supporting Information S3A and S3B. 

Responses to Reviewer 2

Major areas of potential improvement

1. This study provides a valuable approach for designing best practice inspection interventions, and these interventions would play an important role for biosafety management of clinical laboratories and biosafety laboratories. It would improve the compliance with biosafety guidelines through identifying evidence-based suitable interventions, rather than current top-down approach mainly used. There is a need to describe the detailed methods and related results for this approach in the manuscript for wider application in actual life.

>>> Thanks for your comments. In this version, we have made improvements in describing the methods and related results. More specifically, we improved the descriptions of the LCM (Lines 197-209) and the heterogeneous responses of Classes A and B (Lines 273-274). We also illustrate how customized interventions may be implemented based on our results (Lines 443-456).

Minor areas of potential improvement

1. Line 48: “to aid physicians in in the diagnosis…”, please delete the extra “in”.

>>> Thank you for your suggestion. The extra “in” has been removed from the sentence.

2. About Class A and Class B: The description for them remains unclear. It may be easier to understand in Method section, such as “In other words, the LCM divided the respondents into subgroups, such as Class A and Class B, based on their similar response patterns while also maximizing the differences between these subgroups.” (Line 224-226). Or these subgroups can be tried to explain in Results section: According to the LCM, two subgroups, Class A and B, were identified based on their similar characteristics related to their safety guidelines compliance intentions, where …… (Line 295-297). Please consider seriously and make an appropriate revision.

>>> Thank you for your feedback. We have made the necessary revisions in accordance with your suggestion. Specifically, in the Method section (Lines 208-209), we have added an additional explanation of Class A and Class B as subgroups identified through Latent Class Modeling (LCM) as follows:

In other words, the LCM divided the respondents into subgroups, based on their similar response patterns while also maximizing the differences between these subgroups.

Additionally, in the Results section (Lines 273-275), we have further clarified the characteristics of Class A and Class B in relation to their safety guidelines compliance intentions as follows:

Using the LCM, two subgroups (Class A and Class B) of the respondents with distinct responses to safety guidelines were identified, where Class A represented 31.91% of the cohort, and Class B was the remaining 62.09%

3. Some important results could be described more clearly like S2, such as “For class A, this inspection configuration could yield compliance at 85.48%. (Line 343-344)”,“With this intervention configuration, class B compliance could be as high as 81.40%. (Line 350-351)”, Line 360-364, Line 414-415, Line 281-283 etc. After all, there is a need to explain clearly to a wider audience how these data were obtained, and it may be better to present these results in a table rather than just words.

>>> Thank you for your valuable feedback. To provide a clearer understanding for a wider audience, we have provided a detailed explanation of how each probability was calculated in Supporting Information S3A using an example. Furthermore, in Supporting Information S3B, we provide a spreadsheet that contains the calculation of the probabilities of all 512 scenarios for the homogeneous, Class A, and Class B cases. See Lines 253-256:

We show in detail how the probability analysis was conducted using the baseline as an example in Supporting Information S3A. In Supporting Information S3B, we provide a spreadsheet showing how the probabilities of all 512 intervention scenarios were calculated.

To help the reader to understand how the probability of a mentioned scenario is calculated, we refer to the intervention scenario number in S3B for the reader to check. The following is one example in Lines 262-264. 

The projected compliance of this best practice intervention could improve the probability of compliance from 25.86% at the baseline case (Homogeneous Scenario # 186 in S3B) to 74.54% (Homogeneous Scenario # 488 in S3B). 

4. The author mentions the “baseline” several times in the manuscript, but it remains unclear what it is. It is necessary to explain it or give an appropriate description.

>>> Thank you for your feedback. The baseline case is now defined clearly. Please see a new paragraph in Lines 216-220 in the manuscript, which is shown below.

In DCEs, each scenario was analyzed against a baseline case, which referred to the scenario where all attributes were set to their reference levels as defined in Table 1. This includes the laboratory safety inspector being the lab director, a weekly inspection frequency with a fixed day and time, no feedback on an inspection outcome, and no consequences irrespective of the inspection outcome.

---

## [Decision Letter · Decision Letter 2]

3 Oct 2023

Best Practices for Implementing Biosafety Inspections in a Clinical Laboratory: Evidence from a Multi-Site Experimental Study

PONE-D-22-26396R2

Dear Dr. Chen,

We’re pleased to inform you that your manuscript has been judged scientifically suitable for publication and will be formally accepted for publication once it meets all outstanding technical requirements.

Kind regards,

Ehsan Kamrani

Academic Editor

PLOS ONE
---

## [Editor Report · Acceptance letter]

5 Oct 2023

PONE-D-22-26396R2 

Best Practices for Implementing Biosafety Inspections in a Clinical Laboratory: Evidence from a Multi-Site Experimental Study 

Dear Dr. Chen:

I'm pleased to inform you that your manuscript has been deemed suitable for publication in PLOS ONE. Congratulations! Your manuscript is now with our production department. 

Kind regards, 

on behalf of

Dr. Ehsan Kamrani 

Academic Editor

PLOS ONE